# Comprehensive Functional Annotation of Metagenomes and Microbial Genomes Using a Deep Learning-Based Method

Mary Maranga,[a] Pawel Szczerbiak,[a] Valentyn Bezshapkin,[a] Vladimir Gligorijevic,[b,c] Chris Chandler,[b] Richard Bonneau,[b,c] Ramnik J. Xavier,[d,e,f,g] Tommi Vatanen,[d,h,i] Tomasz Kosciolek[a]

[a]Malopolska Centre of Biotechnology, Jagiellonian University, Krakow, Poland

[b]Center for Computational Biology, Flatiron Institute, Simons Foundation, New York, New York, USA

[c]Prescient Design, New York, New York, USA

[d]Broad Institute, Cambridge, Massachusetts, USA

[e]Center for Microbiome Informatics and Therapeutics, MIT, Cambridge, Massachusetts, USA

[f]Center for Computational and Integrative Biology, Department of Molecular Biology, Massachusetts General Hospital, Harvard Medical School, Boston, Massachusetts, USA

[g]Klarman Cell Observatory, Broad Institute of MIT and Harvard, Cambridge, Massachusetts, USA

[h]Liggins Institute, University of Auckland, Auckland, New Zealand

[i]Research Program for Clinical and Molecular Metabolism, Faculty of Medicine, University of Helsinki, Helsinki, Finland

**ABSTRACT** Comprehensive protein function annotation is essential for understanding microbiome-related disease mechanisms in the host organisms. However, a large portion of human gut microbial proteins lack functional annotation. Here, we have developed a new metagenome analysis workflow integrating *de novo* genome reconstruction, taxonomic profiling, and deep learning-based functional annotations from DeepFRI. This is the first approach to apply deep learning-based functional annotations in metagenomics. We validate DeepFRI functional annotations by comparing them to orthology-based annotations from eggNOG on a set of 1,070 infant metagenomes from the DIABIMMUNE cohort. Using this workflow, we generated a sequence catalogue of 1.9 million nonredundant microbial genes. The functional annotations revealed 70% concordance between Gene Ontology annotations predicted by DeepFRI and eggNOG. DeepFRI improved the annotation coverage, with 99% of the gene catalogue obtaining Gene Ontology molecular function annotations, although they are less specific than those from eggNOG. Additionally, we constructed pangenomes in a reference-free manner using high-quality metagenome-assembled genomes (MAGs) and analyzed the associated annotations. eggNOG annotated more genes on well-studied organisms, such as *Escherichia coli*, while DeepFRI was less sensitive to taxa. Further, we show that DeepFRI provides additional annotations in comparison to the previous DIABIMMUNE studies. This workflow will contribute to novel understanding of the functional signature of the human gut microbiome in health and disease as well as guiding future metagenomics studies.

**IMPORTANCE** The past decade has seen advancement in high-throughput sequencing technologies resulting in rapid accumulation of genomic data from microbial communities. While this growth in sequence data and gene discovery is impressive, the majority of microbial gene functions remain uncharacterized. The coverage of functional information coming from either experimental sources or inferences is low. To solve these challenges, we have developed a new workflow to computationally assemble microbial genomes and annotate the genes using a deep learning-based model DeepFRI. This improved microbial gene annotation coverage to 1.9 million metagenome-assembled genes, representing 99% of the assembled genes, which is a significant improvement compared to 12% Gene Ontology term annotation coverage by commonly used orthology-based approaches. Importantly, the workflow supports

Address correspondence to Tomasz Kosciolek, tomasz.kosciolek@uj.edu.pl, or Tommi Vatanen, tommi.vatanen@helsinki.fi.

The authors declare no conflict of interest.

pangenome reconstruction in a reference-free manner, allowing us to analyze the functional potential of individual bacterial species. We therefore propose this alternative approach combining deep-learning functional predictions with the commonly used orthology-based annotations as one that could help us uncover novel functions observed in metagenomic microbiome studies.

**KEYWORDS** genome, metagenome, orthology, pangenome, deep learning, functional annotation, gene function, metagenome-assembled genomes, metagenomics, microbiome

The advent of sequencing technologies has resulted in a substantial increase in metagenomic studies sequencing DNA of microbial consortia (microbiomes) inhabiting various hosts and environments. This has led to a significant boost in known microbial genomes, contributing substantially to our understanding of the genetic diversity encoded in microbiomes. Although metagenomics provides a great potential to characterize difficult-to-cultivate and uncultivated microbes, our ability to link specific genes to disease phenotypes is hampered by poor understanding of the functions and roles of the majority of the newly discovered microbial genes (1, 2).

Comprehensive functional annotation is crucial in identifying disease-causing functional changes in proteins, detecting antibiotic resistance genes (3, 4), and designing new therapeutic strategies. Labor-intensive laboratory experiments still provide the most reliable means of functionally annotating genes and the proteins they encode (5). However, due to the rapid increase in the number of uncharacterized proteins, such experimental methods and manual curation fail to scale up to accommodate such a large amount of protein sequence data. This has created a huge sequence-to-function gap which is still widening, because sequencing is high throughput, while functional characterization continues to be relatively slow. The overall proportion of human gut microbiome protein-coding genes with uncharacterized functions ranges between 40 and 60% depending on the annotation method (1, 6). A recent study reported that 27.3% of genes present in the Unified Human Gastrointestinal Genome (UHGG) catalogue could not be mapped to functional databases, while 14.2% of genes matching the Clusters of Orthologous Groups (COG) database were labeled as "function unknown" (7). Automated and scalable methods for microbial protein function prediction are needed to address this gap.

Most functional annotation methods rely on inferring homology across databases such as UniProt (8) and the NCBI's reference sequence (RefSeq) database (9). Conventional homology-based annotation methods are fast; however, they suffer from low functional annotation coverage. Deep learning methods have been considered an effective complement (10–12), since they are able to predict protein functions on a large scale, irrespective of sequence homology. Metagenome functional annotation can be performed either by assembling sequence reads into contigs, followed by reconstruction of complete and accurate metagenome-assembled genomes (MAGs) and mapping predicted genes to annotated sequences (13), or by directly mapping individual reads to annotated gene sequences (e.g., see references 14–16). Various read mapping-based functional annotation workflows exist, including HUMAnN2 (15), HUMAnN3 (17) MG-RAST (18), and MetaLAFFA (19). These workflows align short reads directly to a reference sequence catalogue to estimate functional profiles. This approach is, however, limited and fails to annotate genes that lack a homologous counterpart in the reference collection. Sensitive profile-based metagenomics functional prediction tools such as METABOLIC (20) have also been implemented. METABOLIC takes as input genome sequences and queries them against hidden Markov model (HMM)-based databases such as Kofam (21), TIGRfam (22), and custom metabolic HMM profiles.

Reconstruction of MAGs has proved to be a successful strategy for novel genome discovery and functional characterization of complex microbial communities (7). The sequencing reads are assembled into long contiguous DNA fragments and further clustered into different bins based on depth of coverage and tetranucleotide frequency

across samples (23, 24). Several studies have employed this technique, providing new insights into the genetic diversity of the human gut microbiome and paving ways for exploring microbial dark matter (25, 26). Despite these developments, it is difficult to determine the accuracy of metagenomics function predictions, as the majority of the protein functions lack experimental validation. For example, in databases such as UniProtKB (The UniProt Consortium), which contains over 100 million protein sequences, only 1% of the proteins have experimental Gene Ontology (GO) annotations (27, 28).

Widely used schemes for classifying protein functions include GO (29), Enzyme Commission (EC) numbers (30), and the Kyoto Encyclopedia of Genes and Genomes (KEGG) (31). Gene ontology (GO) terms are attractive, as they offer an accurate description of the protein functions, and relationships between those annotations. The system represents function in a directed acyclic graph (DAG)-type relationship, where protein attributes are divided into three main categories: molecular function (MF), biological process (BP), and cellular component (CC) (29). Importantly, GO terms facilitate more comprehensive intermethod comparisons through these protein annotation relationships even when the specificity of annotations differs between methods.

Comprehensive deep learning-based functional annotation has been applied in different research fields, such as human genomics. However, its application in metagenomics is lagging. We addressed this gap by developing a new metagenome assembly workflow integrating *de novo* genome reconstruction, taxonomic profiling, and deep learning-based gene annotations from DeepFRI (10). To the best of our knowledge, this is the first approach in metagenomics to use deep learning-based functional annotations. We validated the functional annotations by comparing them to orthology-based annotations from eggNOG (32). We further show that such an approach almost exhaustively annotates millions of metagenome-assembled genes, although the annotations are, on average, less specific than the homology-based annotations from eggNOG. Finally, we demonstrate integration of gene annotations and metagenome-assembled pangenomes using >1,000 infant metagenomes from the DIABIMMUNE cohort. This workflow contributes to the existing metagenomics pipelines by supporting pangenome reconstruction in a reference-free manner, allowing tracking of shared genes in species. Furthermore, we show that DeepFRI provides additional annotations in comparison to the previous DIABIMMUNE analysis. In summary, combining gene annotations from DeepFRI with the commonly used orthology-based annotations helps with understanding the roles of novel microbial genes observed in metagenomic surveys and could eventually close the sequence-to-function gap hindering most microbiome studies.

## RESULTS

**Workflow architecture.** The workflow is fully automated for metagenomic assembly, binning of metagenome-assembled genomes, construction of gene catalogue, and functional annotation. It integrates state-of-the-art bioinformatic tools via Docker containers. The workflow is implemented using Workflow Description Language (WDL) and allows flexibility of bioinformatic tool versioning and scalability of memory in high-performance computing environments. This allows large-scale functional annotations of metagenomics data by leveraging high-quality protein information to annotate functions with higher coverage. It takes raw paired-end Illumina reads (short reads) as input and performs data analysis in five phases: (i) assembly of sequencing reads into contigs, gene prediction, and clustering to generate a gene catalogue, (ii) functional annotation of the gene catalogue, (iii) binning of assembled contigs into MAGs, (iv) taxonomic annotation of MAGs, and (v) mapping between MAGs and functionally annotated gene catalogue (Fig. 1). We have developed a custom Python script for mapping between metagenomic species and the functionally annotated gene catalogue. On completion, the workflow provides various output files, such as the MAGs constructed, nonredundant gene catalogue, functional annotations, and gene mapper table as tab-separated output files.

**Undescribed diversity in infant gut metagenomes.** To demonstrate feasibility and robustness and to highlight several uses of the workflow data products, we analyzed

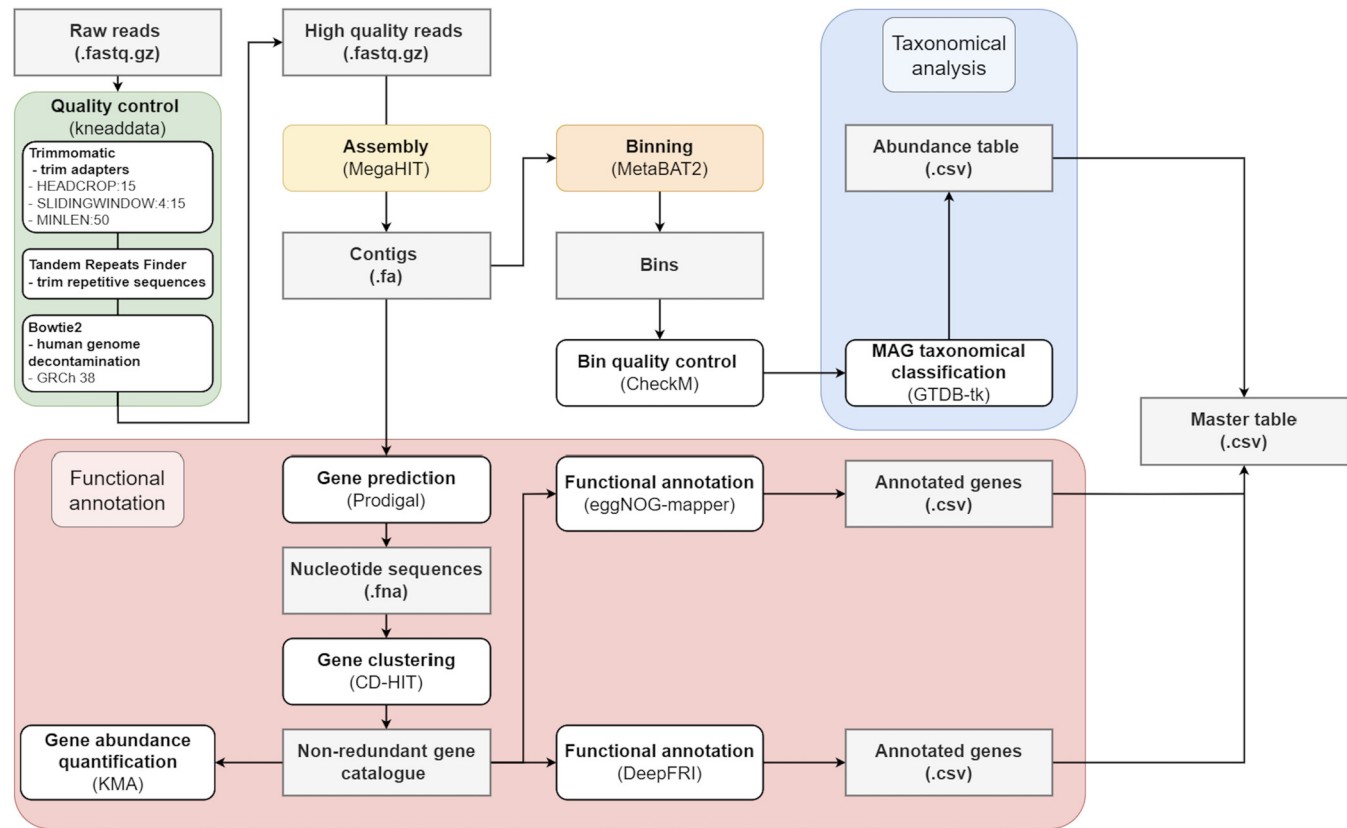

**FIG 1** Schematic workflow overview.

metagenomic data from the DIABIMMUNE cohort (33) (https://diabimmune.broadinstitute.org/diabimmune/). This constituted a longitudinal sample series from individual infants and young children in Finland, Estonia, and Russia. Overall, we aggregated metagenome shotgun sequencing data from 1,070 samples, including 202 samples from Estonia, 586 samples from Finland, and 282 samples from Russia. The average number of reads per sample was $1.49 \times 10^7$ paired-end reads (minimum, $9.28 \times 10^3$; maximum, $8.09 \times 10^7$).

The first step for functional analysis was the construction of a nonredundant gene catalogue. We performed *de novo* assembly of the metagenomes, resulting in a total of 17 million long contigs with lengths of ≥500 bp and an average of 16,510 contigs per sample (minimum, 94 contigs; maximum, 60,979 contigs), collectively harboring approximately 21.9 million open reading frames (ORFs), as predicted by Prodigal. Clustering of these genes into gene families with >95% sequence similarity resulted in a catalogue of 1.9 million nonredundant genes.

Metabat binning of contigs produced a collection of 7,174 MAGs, of which 2,255 were nearly complete bacterial genomes (≥90% completeness, <5% contamination), which corresponded to 2 nearly complete MAGs per sample, on average. Taxonomic annotation of each genome was carried out using GTDB-tk (34). The high-quality genomes spanned 202 bacterial species, with most MAGs representing the phyla *Firmicutes* (884 genomes), *Bacteroidota* (823 genomes), and *Actinobacteriota* (296) and the genera *Bacteroides* (614 genomes), *Bifidobacterium* (268 genomes), and *Faecalibacterium* (124 genomes) (see Fig. S1a and b in the supplemental material).

To visualize the distribution of high-quality, nearly complete genomes across phylogeny, we built a maximum-likelihood phylogenetic tree based on high-quality, nearly complete genomes. The tree was constructed with PhyloPhlAN (35) and visualized using Interactive Tree of Life (v5.6.2) (36). The genomes spanned 10 bacterial classes, including *Coriobacteriia*, *Actinomycetia*, *Bacilli*, *Negativicutes*, *Clostridia*, *Campylobacteria*, *Alphaproteobacteria*, *Gammaproteobacteria*, *Verrucomicrobiae*, and *Bacteroidia* (Fig. 2a).

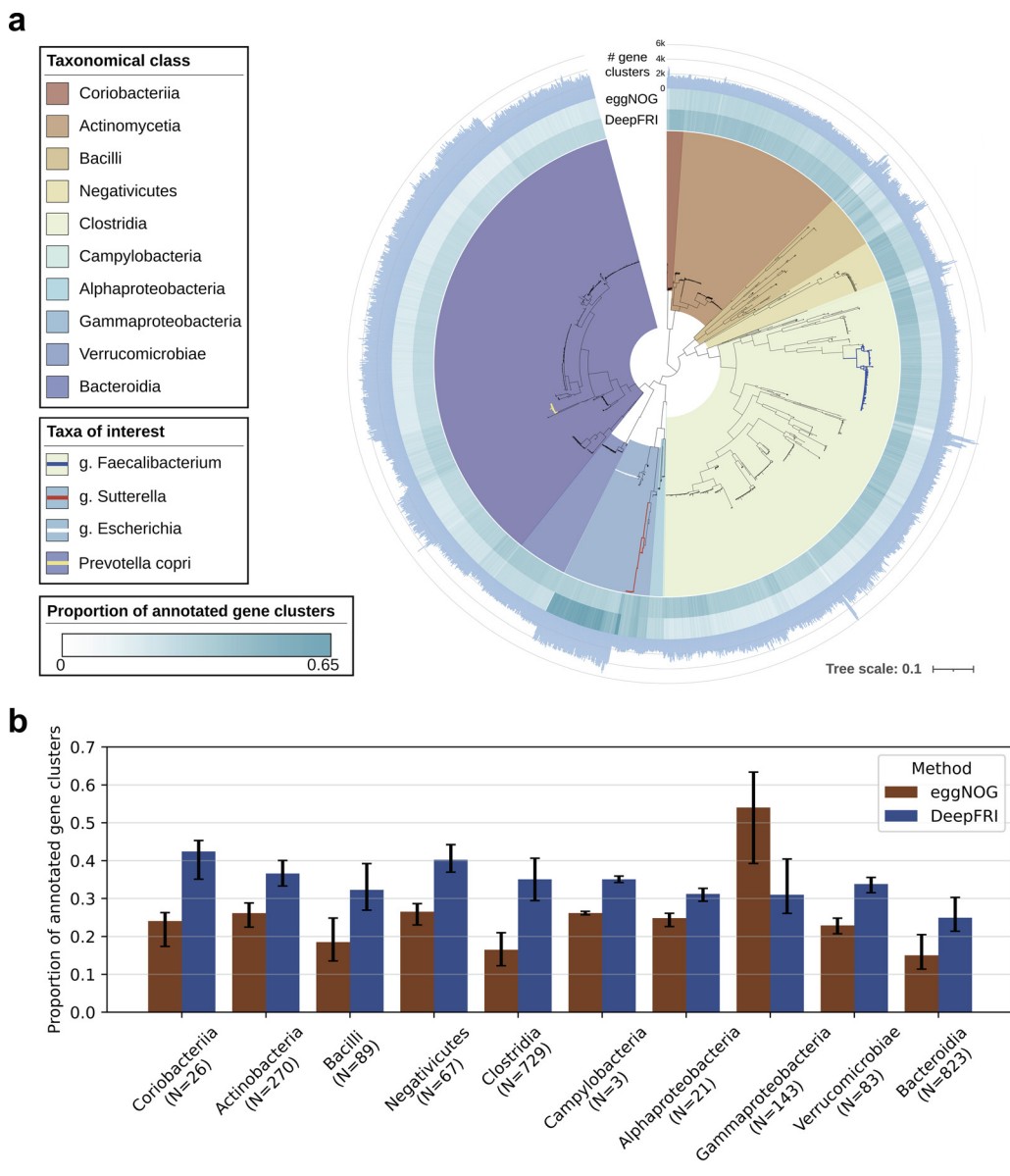

**FIG 2** Phylogenetic analysis and taxonomic annotation of high-quality metagenome-assembled genomes. (a) Maximum-likelihood phylogenetic tree of 2,255 high-quality, nearly complete genomes. The taxonomy of the MAGs was assigned by GTDB-Tk. The innermost layer corresponds to 10 bacterial classes. The second and third rings represent the proportions of genes annotated by DeepFRI and eggNOG. Bars in the outermost layer indicate the number of gene families per MAG. (b) Distribution of annotated genes per class (bar plot shows mean proportion of annotated genes, and error bars show 5th and 95th percentiles of proportions within a taxonomical class on the x axis). "N" refers to the number of genomes in each class.

We observed high phylogenetic dispersion in the genus *Faecalibacterium*. *Faecalibacterium* has been shown to be highly diverse and to comprise multiple phylotypes (2).

To integrate information on gene homology from the gene catalogue and taxonomic information obtained through MAGs, we tried to provide further insights into the following question: are 95% gene families taxon specific or shared across genomes of different species, genera, or even families? We selected only MAGS with a contamination threshold of <5% and calculated the number of gene families assigned with multiple taxonomies. Interestingly, we found that a notable portion of the gene families had multiple taxonomies assigned, 32,156 (3%) at the family level, 92,355 (8%) at the genus level, and 179,355 (15%) at the species level (Fig. S1c). Although a part of

these observations could be explained by highly conserved genes with distant common ancestors, we hypothesize that a portion of the shared genes reflect more recent horizontal gene transfer events. Additional studies with more precise genome reconstruction methods, such as through long-read sequencing and/or isolate strain sequencing, could further quantify the prevalence of such gene transfer events.

**Functional representation of the DIABIMMUNE infant metagenomes.** To elucidate the functional composition of the gut microbiota, we annotated the nonredundant gene catalogue (1.9 million genes) using DeepFRI and eggNOG. DeepFRI predictions were defined in the following manner: DeepFRI scores of ≥0.2 (standard quality) and DeepFRI scores of ≥0.5 (high quality). For comparisons between eggNOG and DeepFRI, we focused on the molecular function branch of Gene Ontology and considered DeepFRI annotations with a threshold above 0.2 to be significant. We first validated the predicted annotations by comparing the level of concordance between both methods based on the specificity of the GO terms, expressed as Shannon information content. The information content (IC) is used to quantify the specificity of a GO term in the context of the entire set of annotations, such that GO terms annotating many genes are considered general and hence contain low information content, while rarely occurring GO terms are considered more specific and contain higher information content (IC values) (37). We propagated the Gene Ontology annotations upward through the child-parent relationships using a list of information content values obtained from reference 10, in which IC was calculated as $-\log_2\text{Prob}(\text{GO}_i)$, where $\text{Prob}(\text{GO}_i)$ is the probability of GO term $i$ occurring in the UniProt-GOA database ($n_i/n$, where $n_i$ is the number of annotations with a term $i$ and $n$ is the total number of annotated proteins in UniProt-GOA) (10).

By comparing consensus in the Gene Ontology annotations predicted by each method, we observed that DeepFRI annotated more genes, although the annotations are, on average, less specific than those obtained with eggNOG (Fig. 3a). DeepFRI predicted an order of magnitude more low-information-content gene functions (information content between 1 and 7) (see Table 1 at https://github.com/bioinf-mcb/metagenome_assembly/tree/master/supplementary_tables). Previous DIABIMMUNE microbiome analysis relied on a homology-based functional search in HUMAnN2 (33). We further included HUMAnN2 Gene Ontology annotations to this comparison. HUMAnN2 predicted higher-information-content GO terms to a larger number of genes' MF-GO terms (IC > 12) (Fig. 3a).

Next, we wanted to assess whether genes annotated by both methods gained function at the same information content level. We further stratified annotations into different categories: (i) concordant annotations, where eggNOG and DeepFRI annotations agree (genes obtained functions at the same information content level); (ii) discordant, where eggNOG and DeepFRI annotations disagree but annotations from both methods exist; (iii) DeepFRI unique (genes obtained function from only this method); and (iv) eggNOG unique. The concordance between the methods revealed that DeepFRI predictions agree well with eggNOG predictions, with an average of 70% concordance (Fig. 3b). The concordance between DeepFRI and HUMAnN2 was 86% on average (Fig. 3c). An average of 84% of the annotations are concordant between eggNOG and HUMAnN2 (Fig. S2b). We observed an increase in concordant annotations between eggNOG and HUMAnN2 at higher information content levels, i.e., >12. On the other hand, 16% of the annotations were discordant even though both methods are based on sequence homology. The similar levels of concordance between DeepFRI and eggNOG/HUMAnN2 support the reliability of DeepFRI predictions. Additionally, DeepFRI had a high number of unique annotations, thus expanding the functional landscape of metagenome-assembled genes (Fig. 3d).

We then looked into functions encoded by the microbial genes. In total, nearly all the gene clusters in the gene catalogue (1,895,540 [99%]) obtained GO molecular function annotations by DeepFRI standard quality (0.2 threshold), whereas only 219,634 (12%) genes obtained similar annotations by eggNOG. Figure S3 shows DeepFRI annotation rate as a function of gene cluster size; we observed that the larger clusters were well annotated. Further, we complemented the annotations by considering functional

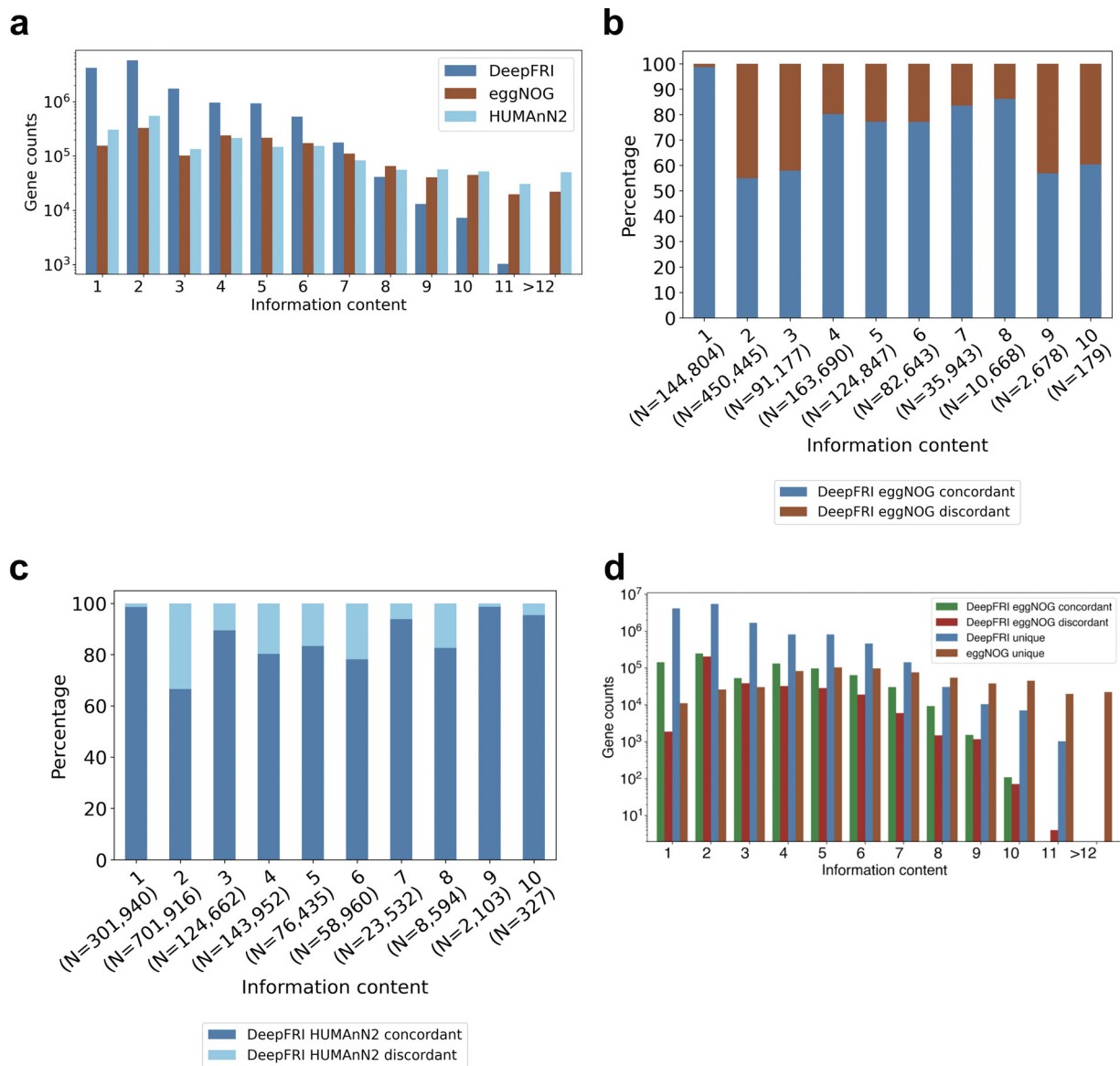

**FIG 3** Concordance between DeepFRI, eggNOG, and HUMAnN2 annotations. (a) The information content of gene functions predictions by DeepFRI, eggNOG, and HUMAnN2. (b) Percentage of concordant and discordant annotations between DeepFRI and eggNOG per each information content level. (c) Percentage of concordant and discordant annotations between DeepFRI and HUMAnN2 per information content level. (d) Consensus between DeepFRI and eggNOG annotations. See Table 1 at https://github.com/bioinf-mcb/metagenome_assembly/tree/master/supplementary_tables for the list of information content values.

annotations from the eggNOG free text description, which consolidates information obtained from several source databases, such as SMART/Pfam (32). A total of 1,372,653 (72%) genes obtained functional information from the eggNOG free text description. We further compared differences in gene sets annotated by both methods. The Venn diagram (Fig. 4a) shows the intersection among genes annotated by DeepFRI and eggNOG, with 219,003 genes in common across the two gene sets. A total of 1.6 million genes were unique to DeepFRI and 631 genes unique to eggNOG gene ontology. Additionally, 1.3 million gene sets were common across DeepFRI and eggNOG free text description annotations (Fig. 4b).

**Filtering informative Gene Ontology terms.** The Gene Ontology protein function description is hierarchical, such that a protein can have multiple GO terms annotated, and a term can have multiple relationships to broader parent terms and more specific child terms (child-parent relationship) (29). For example, the molecular function term

## All Gene Ontology sets

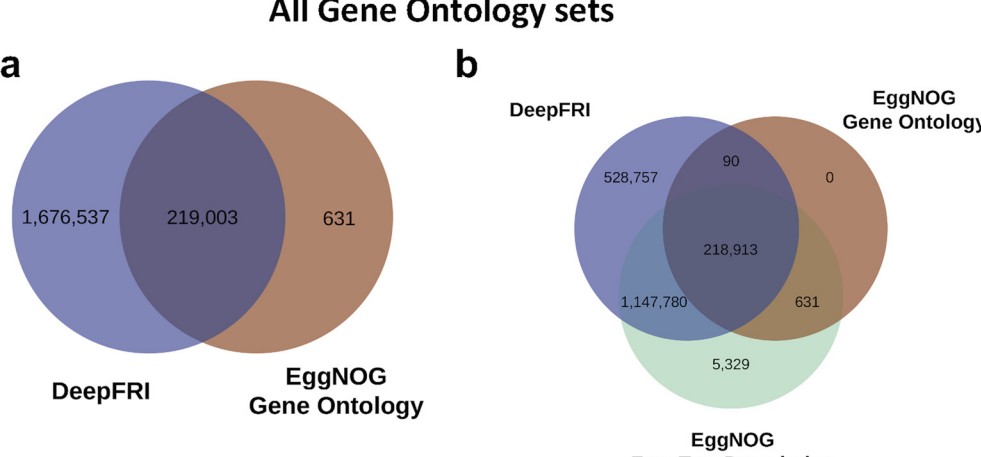

## Informative Gene Ontology sets

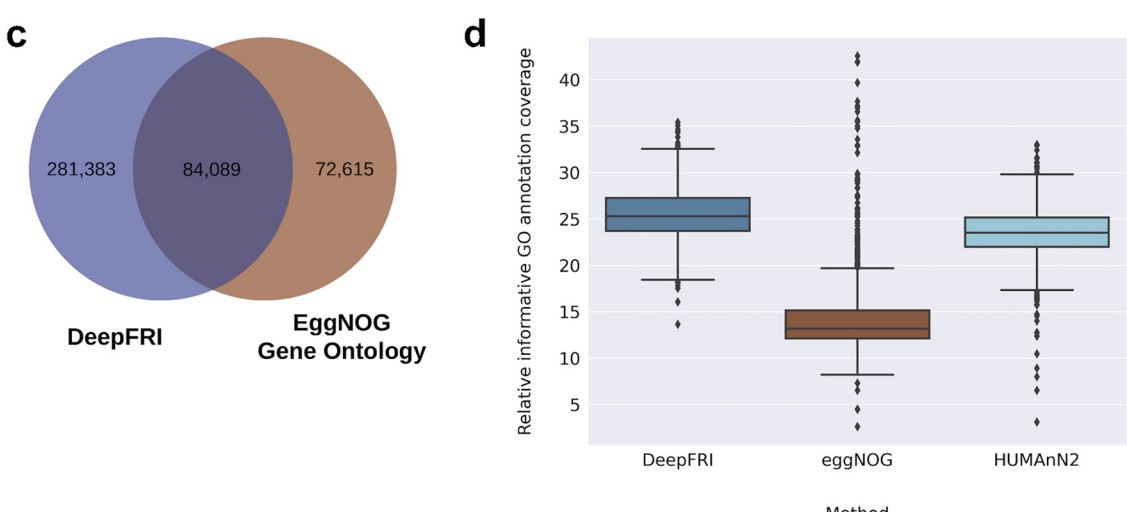

**FIG 4** Comparison of predictions between DeepFRI, eggNOG, and HUMAnN2. (a) Venn diagram of the number of gene sets annotated by DeepFRI and eggNOG gene ontology (all GO terms). (b) Three-way comparisons of gene sets annotated by DeepFRI, eggNOG gene ontology (all GO terms), and eggNOG free text description. (c) Venn diagram comparisons of gene sets annotated by DeepFRI and eggNOG using only informative gene ontology terms. (d) Abundance of genes (in CPM) annotated by DeepFRI and eggNOG and by HUMAnN2 (informative gene ontology terms). The annotation is weighted by relative abundances normalized to copies per million. Annotation rate as a function of cluster size is shown in Fig. S3.

"ATP binding" has the parent term "binding." For downstream analysis, we filtered out the general gene ontology terms using a subset of 614 informative GO terms obtained from previous work by Vatanen et al. (38), following the method proposed in references 39 and 40. The informative GO categories were obtained by traversing the GO tree from the root and selecting terms that satisfy the following parameters: (i) terms were associated with more than 2,000 proteins (*k* value) and (ii) each of their descendant terms contained fewer than 2,000 proteins ($k = 2,000$ equates to approximately 1 of every 5,000 UniRef50 protein families). This informative gene ontology set provides more resolution to extensively studied processes (33). Filtering of gene ontology terms reduced the number of annotated genes by both methods, resulting in 365,472 genes annotated by DeepFRI and 156,704 genes annotated by eggNOG. The Venn diagram (Fig. 4c) shows the intersection of 84,089 genes common between DeepFRI and eggNOG gene ontology. A total of 281,383 genes were unique to DeepFRI, and 72,615 genes were unique to eggNOG.

Finally, we evaluated the relative abundance of annotated genes between DeepFRI, HUMAnN2, and eggNOG considering only genes annotated with informative Gene Ontology sets. We computed the proportion of metagenomic gene abundance with functional annotation as follows: (i) paired-end reads were mapped to the gene catalogue, and (ii) per-gene alignment statistics were then weighted based on the target sequence length to produce abundance values for each gene family. We observed an increase in the annotation coverage at the metagenome level by DeepFRI, with an average coverage of 25.6%, in comparison to HUMAnN2 and eggNOG, which had average coverages of 23.4% and 14.5%, respectively (Fig. 4d). A Wilcoxon signed-rank test was performed to compare the annotations between the methods. This revealed that DeepFRI and eggNOG annotations were statistically different ($P = 4.74 \times 10^{-177}$). Similarly, there was a statistically significant difference between DeepFRI and HUMAnN2 annotations ($P = 2.60 \times 10^{-155}$). The main molecular functions annotated by DeepFRI and eggNOG are ATP binding (GO:0005524), structural constituent of ribosome (GO:0003735), and zinc ion binding (GO:0008270) (see Table 2 at https://github.com/bioinf-mcb/metagenome_assembly/tree/master/supplementary_tables).

We further investigated how DeepFRI performs at a higher threshold (DeepFRI score ≥ 0.5). Figure S4 shows comparisons of concordance between eggNOG and DeepFRI threshold (0.5). We observed an average agreement of 73% between eggNOG and DeepFRI annotations (Fig. S4b). Additionally, a total of (845,897;44%) genes obtained GO annotations. The comparisons between DeepFRI (high-quality threshold) and eggNOG are shown in Fig. S5a to d.

**Variation in GO abundance over time.** Longitudinal trends are ubiquitous in infant microbiome data. We therefore measured longitudinal effects on relative abundance of a subset of GO terms using linear mixed-effect models. For each GO term abundance (summed copies per million [CPM]), a linear mixed-effect model was fitted with age (in months) at collection as a fixed effect and subject ID as a random effect (assuming the same slope and different intercept per subject ID). We identified 8 GO terms with statistically significant ($P < 0.05$) longitudinal trends using DeepFRI annotations and 20 and 13 such GO terms using eggNOG and HUMAnN2 annotations, respectively (Fig. S6a and b). For example, we observed a depletion of genes annotated with ATP binding (GO:0005524) with age for DeepFRI and eggNOG annotations (Pearson $R$ correlations of 0.60 and 0.66, respectively), while HUMAnN2 annotations displayed a different trend (increase in annotated genes over time (Pearson $R = 0.51$) (Fig. S6c). Additionally, there was a depletion of genes annotated with magnesium ion binding function (GO:0000287) with age for the three methods. DeepFRI-annotated GO terms having less longitudinal trends might indicate that part of the trends observed in eggNOG and HUMAnN2 annotations arise from database biases. For example, we previously noted a depletion of annotated genes in microbiomes of newborns compared to microbiomes later in life (6, 41). Although DeepFRI annotations appear to be more speculative by nature, they could also provide a conservative view of longitudinal microbiome trends in infancy.

**Pangenome diversity patterns within the infant metagenomes.** The pangenome can be defined as the entire gene repertoire of all strains in a species (42). Genes in a pangenome are classified into two categories: core and accessory genes. The core genes are shared by genomes within a species, while accessory genes are present in a subset of the genomes within a species. The pangenomes were constructed in the following manner. We recruited nearly complete metagenome-assembled genomes (≥90% completeness and <5% contamination) and included species with at least 10 independent nearly complete MAGs. We then considered the genes present in ≥90% of MAGs of each species core genes (accounting for the incompleteness of the MAGs), while genes present in <90% of MAGS were considered accessory genes. Combined core and accessory genes constituted the pangenome of a species. The pangenomes were significantly smaller than previous pangenomes constructed from the metagenomic species by gene covariance (33), indicating that the current approach was more conservative (Fig. S7a).

The pangenomes covered a total of 42 bacterial species (Fig. S7b) and consisted of a total of 70,997 core genes (on average, 1,690 core genes per pangenome, ranging from 573 to 3,612 genes) and 355,761 accessory genes (on average, 8,470 accessory genes per pangenome, ranging from 1,222 to 27,879 genes) (Fig. S7c and Table S1). Each species had distinctly different core genome and pangenome sizes. Our pangenome analysis showed that *Veillonella parvula* and *Bifidobacterium pseudocatenulatum* contained the smallest core genomes, with 573 and 753 genes, respectively. In contrast, *Escherichia coli* contained the largest core genome (3,612 genes). To investigate differences in intraspecies gene richness, we analyzed the pangenome size in relation to the number of genomes (MAGs). We found a strong correlation between the number of MAGs and the size of the pangenome, with a Pearson correlation *r* value of 0.8 (Fig. 5a). *Bacteroides ovatus* and *Blautia wexlerae* displayed a larger pangenome size than expected by the trend. *Bacteroides* species have been shown to have large pangenome sizes (43). Their remarkable genome plasticity facilitates adaptation to various ecological niches and interaction with the host immune system (44). *Akkermansia muciniphila*, *Bifidobacterium longum* and *Bifidobacterium bifidum* had a smaller pangenome than expected by the trend.

To explore the diversity within the pangenomes, we compared the accessory gene repertoire of each species. We observed a wide variation in the gene content between various species. *Bacteroides* species had the highest number of shared accessory genes (accessory genes that were observed in more than one species) (Fig. 5b). This observation is consistent with a recent study that showed that *Bacteroides* species exchange genes within the genus more frequently through horizontal gene transfer compared to other species such as *Bifidobacterium* (45). Interestingly, accessory genomes of species such as *Veillonella parvula*, *Akkermansia muciniphila*, *Parasutterella excrementihominis*, and *Sutterella wadsworthensis* harbored very low numbers of shared genes, 76, 56, 6, and 1, respectively, and the rest of the accessory genome was unique (Fig. 5b and Table S1). *Bacteroides dorei* is one of the most diverse and prominent members in the infant gut, playing an essential role in immune activation (33, 45). We wanted to look into its genomic diversity by comparing its accessory genome to those of other *Bacteroides* species. The analysis showed that *Bacteroides dorei* harbored 7,514 unique accessory genes (accessory genes not found in any other species), representing 38% of its accessory genome (Fig. S7d). One plausible explanation for this diversity could be gene gain through horizontal gene transfer.

To obtain a better understanding of the functions encoded in the core and accessory genomes, we annotated the pangenomes using DeepFRI and eggNOG. Here, we used only informative GO terms. There was a noticeable difference between annotations predicted by DeepFRI and eggNOG. Our results show that eggNOG annotated more functions on well-studied genomes, such as *Escherichia coli*. The core genome of *Escherichia coli* had the highest annotation level: 1,449 (40%) of core genes obtaining GO term annotations from eggNOG and 880 (24%) from DeepFRI (Fig. 5c). This could be attributed to the fact that *E. coli* is an extensively studied model organism (46). Moreover, this observation was supported by our phylogenetic analysis that showed that eggNOG annotated more genes belonging to *Gammaproteobacteria* (Fig. 2b). On the other hand, DeepFRI annotations were less sensitive to taxa, although DeepFRI annotated more genes belonging to members of the genus *Bacteroides*. The core genes were well covered, with a mean of 25.3% of the genes assigned functions by DeepFRI, while accessory genome had a mean of 15.9% annotated genes. *Bacteroides vulgatus* had the highest number of annotated genes (4,383 [7%] compared to eggNOG, which annotated 928 [4.5%] genes) (Fig. 5c). Our analysis provides detailed identification of the functional profile of these species and can guide future studies aimed at uncovering potential mechanisms responsible for the diversity in *Bacteroides*. Pangenome functional annotation at the DeepFRI high-quality threshold is shown in Fig. S8.

## DISCUSSION

Comprehensive functional assignment of metagenomic sequences is crucial for unlocking the microbiome's clinical potential. Here, we provide a stand-alone and fully

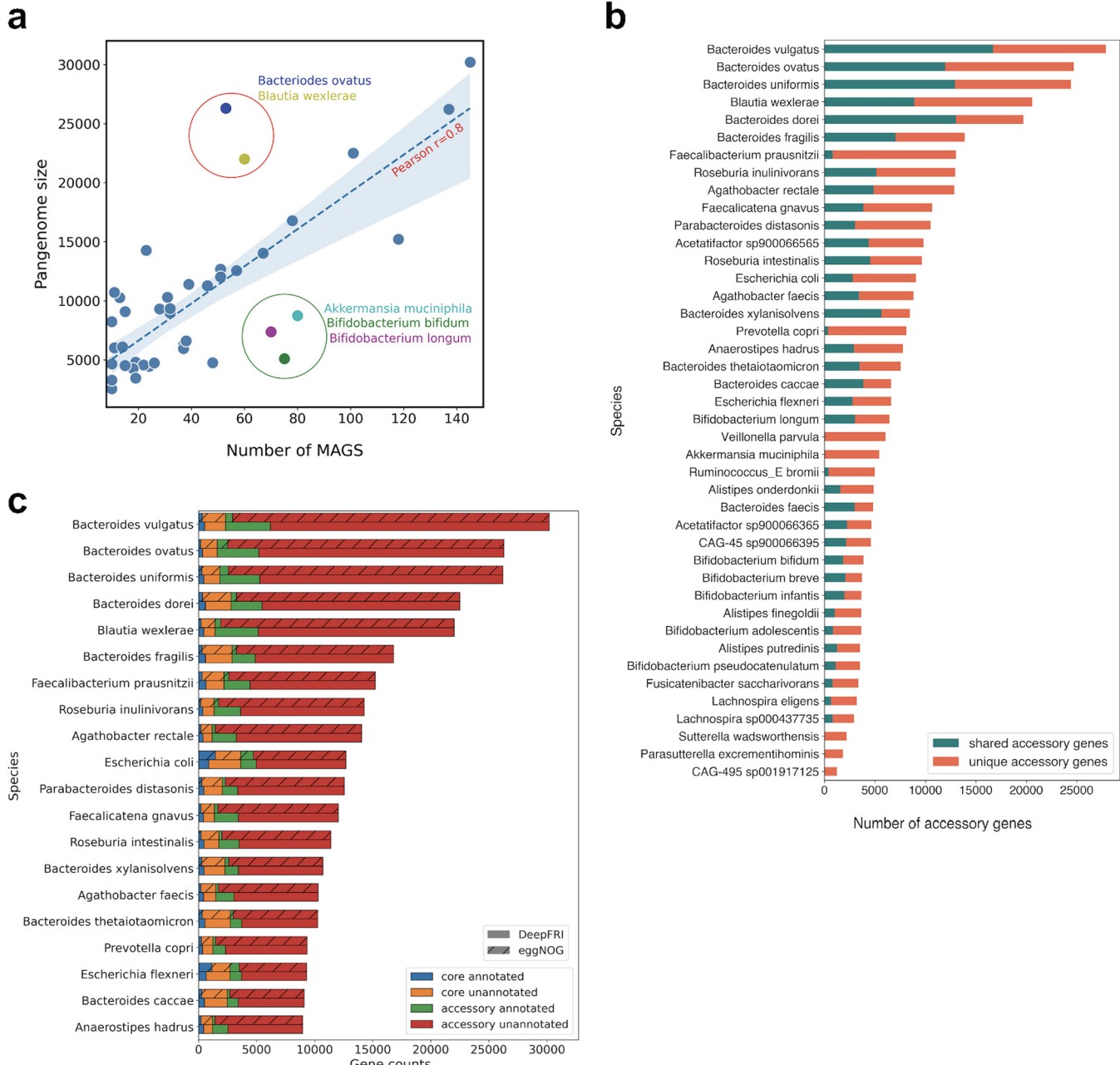

**FIG 5** Pangenome patterns within the infant metagenomes. (a) Pangenome size in relation to the number of genomes (MAGs). (b) Number of unique (not shared with other species) and shared accessory genes per pangenome. See Table S1 for a full pangenome size list. (c) Sizes of the core and accessory genomes per species stratified by the functional annotation of genes using eggNOG and DeepFRI (known versus unknown function). Entries are ordered according to the size of the pangenome (20 of 42 species used to construct the pangenomes). The number of annotated genes was computed using only informative Gene Ontology sets.

automated workflow implementing metagenomic assembly, construction of MAGs, nonredundant gene cataloguing, and comprehensive gene function predictions. The workflow performs different steps of metagenomics analysis in a highly reproducible and scalable environment using Workflow Description Language (WDL). Its features provide functional inference of each gene, including taxonomic classification through better linkage of marker genes in the GTDB reference tree, as well as visualization of nearly complete genomes. This allows robust characterization of the functional potential encoded in the microbial genomes.

To demonstrate the use of our workflow, we tested it on infant gut metagenomes from the DIABIMMUNE cohort (33). The metagenomic single-sample assembly strategy

employed here (47) represents a scalable methodology for large-scale metagenome analysis and profiling of functional diversity within microbial communities. Additionally, the workflow conducts binning of contigs to high-quality genomes using Metabat2 (48). We recovered MAGs with high completeness (≥90%) and low contamination (<5%). Construction of complete or nearly complete MAGs has enabled identification of valuable new genomes from rare species and quantification of intrapopulation diversity (49–51). Additionally, it has also been shown to identify novel genes with distinct functional properties associated with human diseases (23).

For functional annotation, we utilized the advantage of the ability of the deep learning method to process massive amounts of data. DeepFRI ensures efficient and accurate mapping of protein sequences to function annotation on a large scale. It enables discovery of novel protein functions by combining two aspects of information, features learned from protein sequences and contact graphs derived from 3-dimensional structures (10). Combining DeepFRI with the commonly used orthology-based method eggNOG not only improved our annotation capabilities but also helped us understand function better. Moreover, we provide an additional layer of function information obtained from eggNOG, such as Pfam annotations (52).

The functional diversity within the human gut microbiome remains poorly characterized, as most genes still lack functional annotation (7). By leveraging high-quality protein information, we demonstrated that DeepFRI significantly improved the functional annotation coverage of the assembled genes (99%). Further validation of the GO terms predicted by DeepFRI showed a high level of concordance with eggNOG annotations, 70% on average. This indicates that DeepFRI produces reproducible and relevant predictions for the biological interpretation. We show that despite DeepFRI giving less specific (generic) GO term annotations, it provides a more comprehensive functional landscape of almost all metagenome-assembled genes. This increase in annotation coverage represents a key step toward novel understanding of the human gut microbiome, thus alleviating the existing sequence-to-function knowledge gap. Moreover, we provide an important functional resource for future interventions leveraging the gut microbiome to improve health.

Another critical aspect of our workflow is the full support of pangenome analysis. Pangenome analysis provides an undiscovered wealth of information for studying the diversity within species. Here, we provide a way to construct pangenomes in a reference-free manner from nearly complete metagenome-assembled species (≥90% completeness and <5% contamination). MAGs serve as an important resource in pangenome analyses, especially where unculturable species or species without reference genomes are being studied (7). The constructed pangenomes were of high quality, allowing us to identify large number genes shared between the most highly abundant gut commensals, i.e., *Bacteroides* species. Moreover, the functional predictions generated from the species pangenomes revealed a striking difference between DeepFRI and eggNOG annotations. Notably, DeepFRI annotated more functions coming from members of *Bacteroides* species. The pangenome analysis done here could be leveraged for further studies aimed at understanding species acquired functions of biomedical relevance.

It should be noted that this workflow has some limitations. First, we tested the workflow only on infant gut metagenomes. The comparisons performed here may not be generalizable to other microbial communities. Therefore, more robust evaluation is required to demonstrate the performance on complex metagenomes, such as human skin metagenomes, environmental microbiomes (seawater/soil metagenomes), and other specific communities. Second, DeepFRI provides only Gene Ontology annotations; therefore, it is difficult to profile antibiotic resistomes (antibiotic-resistant genes) and genes encoding carbohydrate-active enzymes (CAZymes) (53).

Given the potential and limitations highlighted above, future integrations of high-quality structure information coming from DeepFRI graph convolutional network (GCN) predictions will allow large-scale functional annotation of metagenomics data with

higher accuracy. Additionally, we plan to incorporate KEGG orthology (31) and evaluate genes and pathways associated with disease.

In conclusion, we show that the workflow is robust and reproducible for large-scale metagenomics functional annotation, where short-read sequences can be fully processed into annotated files for downstream analysis and visualization. This workflow contributes to the existing pipelines by combining the strength of deep learning-based and orthology-based functional annotation, thus annotating millions of metagenome-assembled genes. Additionally, the workflow supports reference genome-free pangenome construction, allowing us to uncover shared genes between species. The workflow is available as a Docker image and makes use of standard tools for metagenome analysis and functional annotation. Implementation of the workflow in WDL allows extensive parallelization using high-performance computing and cloud computing environments. Additionally, its modular architecture setup enables tool versioning while allowing users to tweak different parameters to suit their specific needs. The Docker images used in the workflow are publicly available at https://github.com/bioinf-mcb/metagenome_assembly.

## MATERIALS AND METHODS

**Quality control and assembly.** Metagenomics data contain sequencing artifacts such as low-quality reads, host contamination and adapter sequences, which compromise downstream analysis. The quality control phase utilizes kneadData (https://huttenhower.sph.harvard.edu/kneaddata/). KneadData combines Trimmomatic for adapter sequence and low-quality read trimming (54), Bowtie2 (55) for removal of host-derived reads, and Tandem Repeat Finder for removing tandem repeats in DNA (56).

After the quality control process, the workflow uses MegaHIT (v2.4.2) (47) to assemble each sample individually into contigs. MegaHIT has been shown to reconstruct known genomes accurately by employing data structures known as the de Bruijn graph, which decomposes reads into k-mers, reducing the memory requirements (13, 57). Additionally, it captures more genes, allowing higher resolution of functional profiling and phenotype to genotype analysis of specific microbial communities. Assembled contigs less than 500 bp long are filtered out to yield high-quality contigs.

**Construction of the gene catalogue and gene abundance estimation.** For gene annotation, open reading frames from each contig (length $\geq$ 500 bp) are predicted using Prodigal (v2.6.3) (58). The gene products are then clustered at 95% sequence similarity using CD-HIT (v4.8.1) (59) to generate the nonredundant gene catalogue. Thereafter, gene abundance quantification is computed by mapping paired-end reads from each sample against the gene catalogue using the KMA tool (60). Hits to a sequence are weighted based on the target sequence length and further normalized to produce counts in copies per million (CPM).

**Binning of contigs to metagenome-assembled genomes and taxonomic classification.** Construction of metagenome-assembled genomes provides novel biological insights into genetic diversity within the microbial communities (7, 26). Our workflow implements an adaptive binning approach using Metabat2 (v.2.3.0) (48) by clustering contigs into bins (MAGs) based on the use of tetranucleotide frequencies, differential abundance, and coverage across samples. The workflow supports binning for each sample individually, thus recovering a great number of high-quality bins. The minimum contig length used for binning is 500 bp. Moreover, for each MAG, completeness and contamination are determined using single-copy genes with CheckM (v1.0.12) (61). MAGPurify did not improve the results significantly, so we decided not to include it as a part of the pipeline (Fig. S9). Users can then select MAGs of the desired quality for downstream analyses. We selected nearly complete, high-quality genomes as those having $\geq$90% genome completeness and <5% contamination for downstream analysis.

Taxonomic classification is conducted using GTDB-Tk (v2.1.0) (34). GTDB-Tk allows robust taxonomic classification by identifying single-copy marker genes from the Genome Taxonomy Database reference genomes using HMMER (62). The domain-specific marker gene alignments are then concatenated into multiple-sequence alignments and used to construct a reference tree by means of a maximum-likelihood-based phylogenetic inference algorithm. The tool improves the resolution of microbial taxa by classifying MAGs based on their position in the GTDB reference tree, evolutionary divergence, and average nucleotide identity to the reference genomes.

**Functional annotation.** Representative cluster centroids of the gene catalogue are annotated with predicted gene functions using two complementary approaches; deep neural networks in DeepFRI (10), and orthology-based annotation in eggNOG (evolutionary genealogy of genes: nonsupervised orthologous groups), using eggnog-mapper (v2.0.1) (32, 63) (options: -m diamond -d none –tax_scope auto –go_evidence non-electronic –target_orthologs all –seed_ortholog_evalue 0.001 –seed_ortholog_score 60 –query-cover 20 –subject-cover 0). eggNOG provides accurate function prediction by inferring orthology relationships and evolutionary history of proteins. The annotated genes are mapped to various ontologies, thus accurately assigning each gene to its function. These ontologies include Gene Ontology (29), Kyoto Encyclopedia of Genes and Genomes (KEGG) for metabolic pathway analysis (31), COG (64), and SMART/Pfam domains for each group (52, 65).

On the other hand, DeepFRI (10) is a novel deep learning-based function annotation method which applies GCNs to learn and extract features from protein sequences and structures. The method achieves accurate prediction of GO terms in two stages: (i) the first step utilizes long short-term memory (LSTM)-based language model, pretrained on sequences from the Pfam database (66), to extract features from

PDB protein sequences (67), and then (ii) GCN architecture learns structure-function relationships using three graph convolutional layers. DeepFRI has the ability to predict protein functions accurately irrespective of sequence homology and provide interpretability of its predictions via saliency maps. DeepFRI outputs scores for each GO term prediction; it is trained such that scores propagate along the GO tree, with higher scores toward the top of the tree and lower scores at the bottom of the tree. The confidence metric for DeepFRI should be interpreted in the following way: DeepFRI scores of $\geq$0.2 indicate standard quality, while DeepFRI scores of $\geq$0.5 indicate high quality.

**Integration of gene functions, quantification, and metagenome-assembled genomes.** We have developed a custom Python script for generating mapping between functionally annotated nonredundant genes and MAGs. The script takes in a variety of input files: gene catalogue including cluster members, assembled contigs, and MAGs (including taxonomic annotations from GTDB and CheckM quality control information).

Considering that the gene catalogue was created by clustering genes into a representative sequence (centroid) using CD-HIT (59) at a sequence similarity threshold of 95%, we propagated functions within each gene cluster and further annotated genes within MAGs. This establishes a link between microbial genes and species, allowing reconstruction of the functional potential of individual bacterial species, providing a deeper and more comprehensive insight into identification of species-specific genes associated with outcome of interest.

Quantification of individual gene abundances across a given metagenome sample is critical for understanding how variation in the functional composition can impact health as well as in understanding how microbes adapt to various environments (68). We perform quantifications of each gene and then apply the abundance values to quantify gene functions from DeepFRI and eggNOG. To estimate per-sample gene abundances, the quality-controlled reads are aligned directly against the nonredundant gene catalogue using KMA tool (60) with default parameters. The number of reads mapping to each gene is then used as a proxy for its abundance in the sample. For example, the total number of reads mapped to each gene is normalized by the length of the target gene sequence to produce counts in CPM. The relative abundance is calculated as follows: (number of reads mapped to a target gene $\times$ 1,000,000)/total read counts.

MAGs include core genes that have specific and specialized functions as well as accessory genes that are variably present in the genomes. We used MAGs to construct pangenomes in a reference-free manner. To construct a pangenome of a given species, we recruited nearly complete MAGs ($\geq$90% genome completeness and <5%) and species with at least 10 genomes. We then defined genes present in $\geq$90% of MAGs of each species as core genes, while genes present in <90% of MAGS are considered accessory genes. This provides an additional dimension to the analysis of the functional repertoire of core and accessory genomes and gives a glimpse into the functional diversity within species.

**Code availability.** The modular workflow and related Docker images are available at https://github.com/bioinf-mcb/metagenome_assembly. The implementation of the workflow in WDL allows extensive parallelization using Linux-based high-performance computing or cloud computing environments. Detailed information about the program versions used and additional information can be found in the GitHub repository.

## SUPPLEMENTAL MATERIAL

Supplemental material is available online only.
**FIG S1**, TIF file, 3.4 MB.
**FIG S2**, TIF file, 9.6 MB.
**FIG S3**, TIF file, 1.9 MB.
**FIG S4**, TIF file, 3.4 MB.
**FIG S5**, TIF file, 10.5 MB.
**FIG S6**, TIF file, 7.7 MB.
**FIG S7**, TIF file, 8.3 MB.
**FIG S8**, TIF file, 3.4 MB.
**FIG S9**, TIF file, 4.9 MB.
**TABLE S1**, XLSX file, 0.01 MB.

## ACKNOWLEDGMENTS

Prescient Design is a Genentech Accelerator.

M.M., P.S., and T.K. are supported by the Polish National Agency for Academic Exchange grant PPN/PPO/2018/1/00014. T.V. acknowledges the use of the Broad Institute computing facility. The open-access publication of this article was funded by the Priority Research Area BioS under the program "Initiative of Excellence –Research University" at the Jagiellonian University in Krakow.

M.M. performed analyses, contributed analytical tools, and wrote the paper. P.S. contributed analytical tools and performed analyses. V.B. performed analyses and edited the manuscript. V.G. contributed analytical tools. C.C. contributed analytical tools. R.B. supervised the project. R.J.X. contributed analytical tools. T.V. conceived the

study, contributed analytical tools, supervised the project, and wrote the paper. T.K. conceived the study, supervised the project, and wrote the paper.

All authors declare that there is no competing interest.

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
