## [Reviewer comments · mSystems]

Comprehensive functional annotation of metagenomes and microbial genomes using a deep learning-based method

Mary Maranga, Pawel Szczerbiak, Valentyn Bezshapkin, Vladimir Gligorijevic, Chris Chandler, Richard Bonneau, Ramnik Xavier, Tommi Vatanen, and Tomasz Kosciolk

Corresponding Author(s): Tomasz Kosciolk, Uniwersytet Jagiellonski w Krakowie

Review Timeline:

Submission Date:	December 8, 2022
Editorial Decision:	January 6, 2023
Revision Received:	February 3, 2023
Accepted:	February 6, 2023

Editor: Thomas Sharpton

Reviewer(s): The reviewers have opted to remain anonymous.

Transaction Report:

DOI: <https://doi.org/10.1128/msystems.01178-22>

January 6, 2023

Dr. Tomasz Kosciol
Uniwersytet Jagiellonski w Krakowie
Malopolska Centre of Biotechnology
Krakow 30-387
Poland

Re: mSystems01178-22 (Comprehensive functional annotation of metagenomes and microbial genomes using a deep learning-based method)

Dear Dr. Tomasz Kosciol:

Thank you for submitting your manuscript to mSystems. We have completed our review and I am pleased to inform you that, in principle, we expect to accept it for publication in mSystems. However, acceptance will not be final until you have adequately addressed the reviewer comments.

Preparing Revision Guidelines

Sincerely,

Thomas Sharpton

Editor, mSystems

Journals Department
Reviewer comments:

Reviewer #1 (Comments for the Author):

Thank you for addressing my specific points. I have no further requests at this time.

Reviewer #2 (Comments for the Author):

The authors have made significant improvements to the manuscript and documentation. After another read and trying to install/run the tool, I would encourage the authors to consider a separate release of only the DeepFRI annotation functionality that could be more easily incorporated into existing workflows.

Reviewer #3 (Comments for the Author):

I thank the authors for their work addressing the reviewers' concerns. The work is greatly improved and more clearly presents a new metagenomic workflow and interesting methods comparison. I have just a few additional suggestions:

- 1) It is very interesting that some of the differential abundance results change dramatically depending on the annotation method (Fig. S6). It may be worthwhile to highlight some individual examples in the main text.
- 2) There is still no mention of the editor's question of whether any additional IRB approval is needed.
- 3) It would be helpful for the documentation to link to some resources on hardware requirements and a few more details on getting started with Cromwell, as this is not a commonly used tool and could present difficulties for some cluster systems. It would also be helpful to include guidelines on how users should obtain support or report any problems with the pipeline (e.g. GitHub issues or a particular contact person).
- 4) Lines 246-248: It strikes me as misleading to present HGT as the sole most likely hypothesis for why gene families are shared between multiple species/genera. Considering typical ANI distributions, strongly conserved genes could easily be shared between species in the same genus and occasionally genera in the same family.
- 5) Lines 919-920: It should be clarified that this assertion is an assumption - "We assume that microbial protein function is conserved..." As an absolute statement, it's not strictly true, since there are many known examples of single SNPs changing the function of a protein.
- 6) Figure S3 is only mentioned in the Figure 4 legend, not in the main text.
- 7) "Filtering of Gene Ontology terms" is a relatively uninformative section title, it may be better to include something about information content since that is the main focus.
- 8) Line 307-308 is both somewhat confusing and stated overly strongly. It would be better to state "The similar level of concordance...supports the reliability of DEEPfri predictions" or similar.
- 9) There are several sentences with confusing structure and/or minor grammatical errors - the manuscript could benefit from copyediting. For example:
 - "Variation in the GO abundance over time" instead of "Variation in GO term abundances over time"
 - Line 88: range → ranges
 - Line 152: it's → its

Reviewer comments:

Reviewer #1 (Comments for the Author):

Thank you for addressing my specific points. I have no further requests at this time.

Thank you!

Reviewer #2 (Comments for the Author):

R2Q1 The authors have made significant improvements to the manuscript and documentation. After another read and trying to install/run the tool, I would encourage the authors to consider a separate release of only the DeepFRI annotation functionality that could be more easily incorporated into existing workflows.

We thank the reviewer for this important comment. The workflow runs in 11 steps (modules) as follows;

- Pre-processing of reads with Kneaddata
- Metagenomics assembly with Megahit
- Gene prediction
- Mapping of reads against the contigs
- Metagenome binning using MetaBAT2
- Quality assessment of genome bins
- Taxonomic classifications
- Gene clustering with CD-HIT-EST
- Mapping of reads to gene clusters and computing gene counts
- Functional annotation of proteins using both eggNOG-mapper and DeepFRI
- Mapping between MAGs and functionally annotated gene catalogue

For each step, the workflow provides output files that can be analysed independently. For example, the workflow provides DeepFRI functional annotations in tabular output that could be incorporated into existing workflows allowing for

further downstream analysis or comparisons with other functional annotation methods.

We have added clarifications the README file on GitHub.

Reviewer #3 (Comments for the Author):

I thank the authors for their work addressing the reviewers' concerns. The work is greatly improved and more clearly presents a new metagenomic workflow and interesting methods comparison. I have just a few additional suggestions:

R3Q1 It is very interesting that some of the differential abundance results change dramatically depending on the annotation method (Fig. S6). It may be worthwhile to highlight some individual examples in the main text.

We thank the reviewer for this suggestion. We have now included two examples of GO terms showing statistically significant ($p < 0.05$) longitudinal trends, ATP binding (GO:0005524) and Magnesium binding (GO:0000287) (Fig. S6c). We observed a depletion of genes annotated with ATP binding with age for DeepFRI and eggNOG annotations (Pearson R correlation of 0.60 and 0.66 respectively), while HUMAnN2 annotation displayed a different trend (increase in annotated genes over time Pearson R 0.51). Additionally, there was a depletion of genes annotated with magnesium ion binding function with age for the three methods. (line 362-368)

Fig. S6C: Linear mixed effect model fitted to summed CPM annotations against subject age (in months) for two specific GO-terms (upper and lower panels) and different annotation methods (left, middle and right panels). Dots correspond to subject ID (see legend) whereas 'x' markers to LMM fitted values. Blue lines represent the LMM fixed effect regressions. Pearson R correlation coefficient between predicted and fitted values is shown in each panel's title.

R3Q2 There is still no mention of the editor's question of whether any additional IRB approval is needed.

We thank the reviewer for highlighting our omission. We used publicly available data for this study. An IRB approval had been obtained for the Diabimmune cohort (<http://www.diabimmune.org/>). The DIABIMMUNE study was conducted according to the guidelines in the Declaration of Helsinki, and all procedures involving human subjects were approved by the Ethical Committee for Psychiatric Diseases and Diseases in Children and Adolescents, Helsinki and Uusimaa Hospital District (Finland), the Ethics Review Committee on Human Research of the University of Tartu (Estonia) and the Ethical Committee, Ministry of Health and Social Development, Karelian Republic of the Russian Federation (Russia).

Since this data have already been public and published in several research papers, we decided not to include the full IRB information in order to avoid confusion that the data were collected for the purpose of this manuscript.

R3Q3 It would be helpful for the documentation to link to some resources on hardware requirements and a few more details on getting started with Cromwell, as this is not a commonly used tool and could present difficulties for some cluster systems. It would also be helpful to include guidelines on how users should obtain support or report any problems with the pipeline (e.g. GitHub issues or a particular contact person).

We thank the reviewer for this important comment. We have provided documentation about Cromwell usage in the README file. We have also provided guidelines of how the users can report issues as well as details of the contact persons.

R3Q4 Lines 246-248: It strikes me as misleading to present HGT as the sole most likely hypothesis for why gene families are shared between multiple species/genera. Considering typical ANI distributions, strongly conserved genes could easily be shared between species in the same genus and occasionally genera in the same family.

We agree and have modified the sentence to include other possibilities for these observations. The sentence now reads: "Although a part of these observations could be explained by highly conserved genes with distant common ancestors, we hypothesize that a portion of the shared genes reflect more recent horizontal gene transfer events. Additional studies with more precise genome reconstruction methods, such as through long read sequencing and/or isolate strain sequencing, could further quantify the prevalence of such gene transfer events." (lines 214-219)

R3Q5 Lines 919-920: It should be clarified that this assertion is an assumption - "We assume that microbial protein function is conserved..." As an absolute statement, it's not strictly true, since there are many known examples of single SNPs changing the function of a protein.

We agree and have removed the sentence from the revised manuscript.

R3Q6 Figure S3 is only mentioned in the Figure 4 legend, not in the main text.

We have discussed Fig. S3 in the text (lines 291-293)

R3Q7 "Filtering of Gene Ontology terms" is a relatively uninformative section title, it

may be better to include something about information content since that is the main focus.

We thank the reviewer for highlighting this. We have modified the title to read "Filtering informative Gene Ontology terms" (line 304)

R3Q8 Line 307-308 is both somewhat confusing and stated overly strongly. It would be better to state "The similar level of concordance...supports the reliability of DEEPfri predictions" or similar.

We have rectified this in the revised manuscript in line with the comment.

R3Q9 There are several sentences with confusing structure and/or minor grammatical errors - the manuscript could benefit from copyediting. For example: -"Variation in the GO abundance over time" instead of "Variation in GO term abundances over time"

-Line 88: range → ranges

-Line 152: it's → its

We thank the reviewer for pointing out this. We have rectified in the revised manuscript (line 82, 131 and 354)

February 6, 2023

Dr. Tomasz Kosciolk
Uniwersytet Jagiellonski w Krakowie
Malopolska Centre of Biotechnology
Krakow 30-387
Poland

Re: mSystems01178-22R1 (Comprehensive functional annotation of metagenomes and microbial genomes using a deep learning-based method)

Dear Dr. Tomasz Kosciolk:

Thank you for submitting your revised manuscript to mSystems. I appreciate your careful consideration of reviewer comments and am recommending your manuscript for publication in mSystems. Please see the information below about next steps, and congratulations.

Your manuscript has been accepted, and I am forwarding it to the ASM Journals Department for publication. For your reference, ASM Journals' address is given below. Before it can be scheduled for publication, your manuscript will be checked by the mSystems production staff to make sure that all elements meet the technical requirements for publication. They will contact you if anything needs to be revised before copyediting and production can begin. Otherwise, you will be notified when your proofs are ready to be viewed.

If you would like to submit a potential Featured Image, please email a file and a short legend to msystems@asmusa.org. Please note that we can only consider images that (i) the authors created or own and (ii) have not been previously published. By submitting, you agree that the image can be used under the same terms as the published article. File requirements: square dimensions (4" x 4"), 300 dpi resolution, RGB colorspace, TIF file format.

We recognize that the video files can become quite large, and so to avoid quality loss ASM suggests sending the video file via <https://www.wetransfer.com/>. When you have a final version of the video and the still ready to share, please send it to mSystems staff at msystems@asmusa.org.

Sincerely,

Thomas Sharpton
Editor, mSystems

Journals Department
E-mail: mSystems@asmusa.org